

# Ocean acidification of a coastal Antarctic marine microbial community reveals a critical threshold for CO$_2$ tolerance in phytoplankton productivity

Stacy Deppeler[1], Katherina Petrou[2], Kai G. Schulz[3], Karen Westwood[4,5], Imojen Pearce[4], John McKinlay[4], and Andrew Davidson[4,5]

[1]Institute for Marine and Antarctic Studies, University of Tasmania, Private Bag 129, Hobart, Tasmania 7001, Australia
[2]School of Life Sciences, University of Technology Sydney, 15 Broadway, Ultimo, New South Wales 2007, Australia
[3]Centre for Coastal Biogeochemistry, Southern Cross University, Military Rd, East Lismore, NSW, 2480, Australia
[4]Australian Antarctic Division, Department of the Environment and Energy, 203 Channel Highway, Kingston, Tasmania 7050, Australia
[5]Antarctic Climate and Ecosystems Cooperative Research Centre, Private Bag 80, Hobart, Tasmania 7001, Australia

*Correspondence to:* Stacy Deppeler (stacy.deppeler@utas.edu.au)

**Abstract.**

High-latitude oceans are anticipated to be some of the first regions affected by ocean acidification. Despite this, the effect of ocean acidification on natural communities of Antarctic marine microbes is still not well understood. In this study we exposed an early spring, coastal marine microbial community in Prydz Bay to CO$_2$ levels ranging from ambient (343 µatm) to 1641

µatm in six 650 l minicosms. Productivity assays were performed to identify whether a CO$_2$ threshold existed that led to a decline in primary productivity, bacterial productivity, and the accumulation of Chlorophyll $a$ (Chl $a$) and particulate organic matter (POM) in the minicosms. In addition, photophysiological measurements were performed to identify possible mechanisms driving changes in the phytoplankton community. A critical threshold for tolerance to ocean acidification was identified in the phytoplankton community between 953 and 1140 µatm. CO$_2$ levels ≥1140 µatm negatively affected photosynthetic

performance and Chl $a$-normalised primary productivity (csPP$_{14C}$), causing significant reductions in gross primary production (GPP$_{14C}$), Chl $a$ accumulation, nutrient uptake, and POM production. However, there was no effect of CO$_2$ on C:N ratios. Over time, the phytoplankton community acclimated to high CO$_2$ conditions, showing a down-regulation of carbon concentrating mechanisms (CCMs) and likely adjusting other intracellular processes. Bacterial abundance initially increased in CO$_2$ treatments ≥953 µatm (days 3-5), yet gross bacterial production (GBP$_{14C}$) remained unchanged and cell-specific bacterial

productivity (csBP$_{14C}$) was reduced. Towards the end of experiment, GBP$_{14C}$ and csBP$_{14C}$ markedly increased across all treatments regardless of CO$_2$ availability. This coincided with increased organic matter availability (POC and PON) combined with improved efficiency of carbon uptake. Such changes in phytoplankton community production could have negative effects on the Antarctic food web and the biological pump, resulting in negative feedbacks on anthropogenic CO$_2$ uptake. Increases in bacterial abundance under high CO$_2$ conditions may also increase the efficiency of the microbial loop, resulting in increased

organic matter remineralisation and further declines in carbon sequestration.



# 1 Introduction

The Southern Ocean (SO) is a significant sink for anthropogenic $CO_2$ (Metzl et al., 1999; Sabine et al., 2004; Frölicher et al., 2015). Approximately 30% of anthropogenic $CO_2$ emissions have been absorbed by the world's oceans, of which 40% has been
via the SO (Raven and Falkowski, 1999; Sabine et al., 2004; Khatiwala et al., 2009; Takahashi et al., 2009, 2012; Frölicher et al., 2015). While ameliorating $CO_2$ accumulation in the atmosphere, increasing oceanic $CO_2$ uptake alters the chemical balance of surface waters, already decreasing the average pH by 0.1 units since pre-industrial times (Sabine et al., 2004; Raven et al., 2005). If anthropogenic emissions continue unabated, future concentrations of $CO_2$ in the atmosphere are projected to reach $\sim$930 µatm by 2100 and peak at $\sim$2000 µatm by 2250 (Meinshausen et al., 2011; IPCC, 2013). This will result
in a further reduction of the surface ocean pH by up to 0.6 pH units, with unknown consequences to the marine microbial community (Caldeira and Wickett, 2003). High-latitude oceans have been identified as amongst the first regions to experience the negative effects of ocean acidification, causing potentially harmful reductions in the aragonite saturation state and a decline in the ocean's capacity for $CO_2$ uptake in the future (Sabine et al., 2004; Orr et al., 2005; McNeil and Matear, 2008; Fabry et al., 2009; Hauck and Völker, 2015). Marine microbes play a pivotal role in the uptake and storage of $CO_2$ in the ocean,
through phytoplankton photosynthesis and vertical transport of biological carbon to the deep ocean (Longhurst, 1991; Honjo, 2004). As the buffering capacity of the SO decreases over time, the biological contribution to total $CO_2$ uptake is expected to increase in importance (Hauck et al., 2015; Hauck and Völker, 2015). Thus, it is necessary to understand the effects of high $CO_2$ on the productivity of the marine microbial community if we are to predict how they may affect ocean biogeochemistry in the future.

Phytoplankton primary production provides a food source to higher trophic levels and plays a critical role in the sequestration of carbon from the atmosphere into the deep ocean (Azam et al., 1983, 1991; Longhurst, 1991; Honjo, 2004; Fenchel, 2008; Kirchman, 2008). In Antarctic waters it is restricted to a short summer season and is characterised by intense phytoplankton blooms that can reach over 200 mg Chl $a$ m$^{-2}$ (Smith and Nelson, 1986; Nelson et al., 1987; Wright et al., 2010). Relative to elsewhere in the SO, the continental shelf around Antarctica accounts for a disproportionately high percentage of annual
primary productivity (Arrigo et al., 2008a). In coastal Antarctic waters, seasonal $CO_2$ variability can be up to 450 µatm over a year (Gibson and Trull, 1999; Boyd et al., 2008; Moreau et al., 2012; Roden et al., 2013; Tortell et al., 2014). Sea ice forms a barrier to outgassing of $CO_2$ in winter, causing supersaturation of the surface water to $\sim$500 µatm. Intense primary productivity in summer rapidly draws down $CO_2$ to <100 µatm, making this region a significant $CO_2$ sink during summer months (Hoppema et al., 1995; Ducklow et al., 2007; Arrigo et al., 2008b).

Ocean acidification studies on individual phytoplankton species have reported differing trends in primary productivity and growth rates. Increased $CO_2$ enhanced rates of primary productivity (Wu et al., 2010; Trimborn et al., 2013) and growth (Sobrino et al., 2008; Tew et al., 2014; Baragi et al., 2015; Chen et al., 2015; King et al., 2015) in some diatom species, while others have been shown to remain unaffected (Chen and Durbin, 1994; Sobrino et al., 2008; Berge et al., 2010; Trimborn





et al., 2013; Chen et al., 2015; Hoppe et al., 2015; King et al., 2015; Bi et al., 2017). In contrast, $CO_2$-related declines in primary productivity and growth rate have also been observed (Barcelos e Ramos et al., 2014; Hoppe et al., 2015; King et al., 2015; Shi et al., 2017), suggesting that responses to ocean acidification are largely species-specific. These differing responses among phytoplankton species may also cause changes in the composition of phytoplankton communities (Trimborn et al.,

2013). It is difficult to extrapolate the response of individual species to natural communities, as monospecific studies exclude interactions among species and trophic levels. Estimates of $CO_2$ tolerance under laboratory conditions may also be influenced by experimental acclimation periods (Trimborn et al., 2014; Hennon et al., 2015; Torstensson et al., 2015; Li et al., 2017a), differences in experimental conditions (e.g. nutrients, light climate) (Hoppe et al., 2015; Hong et al., 2017; Li et al., 2017b), methods of $CO_2$ manipulation (Shi et al., 2009; Gattuso et al., 2010), as well as region-specific environmental adaptations

(Schaum et al., 2012). Thus, investigations on natural communities are essential in order to fully understand the outcome of these complex interactions.

    The effects of ocean acidification on natural Antarctic phytoplankton communities is currently not well understood (Petrou et al., 2016; Deppeler and Davidson, 2017). Tolerance to $CO_2$ levels up to ∼800 µatm have been reported for natural coastal communities in the West Antarctic Peninsula and Prydz Bay, East Antarctica (Young et al., 2015; Davidson et al., 2016).

Although in Prydz Bay, when $CO_2$ levels exceeded 780 µatm, primary productivity declined and community composition shifted toward smaller, picoeukaryote species (Davidson et al., 2016; Thomson et al., 2016; Westwood et al., submitted). In contrast, Ross Sea phytoplankton communities responded to $CO_2$ levels ≥750 µatm with an increase in primary productivity and abundance of large chain-forming diatoms, suggesting that as $CO_2$ increases in this region, diatoms may increase in dominance over the prymnesiophyte *Phaeocystis antarctica* (Tortell et al., 2008b; Feng et al., 2010). The paucity of information

regarding the ocean acidification response of these Antarctic coastal phytoplankton communities highlights the need for further research to determine region-specific tolerances and potential tipping points in community productivity and composition in Antarctica.

    Bacteria play an essential role in the microbial food web through the remineralisation of nutrients from sinking particles (Azam et al., 1991) and as a food source for heterotrophic nanoflagellates (Pearce et al., 2010). Bacterial populations respond

to increases in phytoplankton primary productivity by increasing their productivity and abundance, with maximum abundance often occurring after the peak of the phytoplankton bloom (Pearce et al., 2007). High $CO_2$ levels have been observed to have either no effect on abundance and productivity (Grossart et al., 2006; Allgaier et al., 2008; Paulino et al., 2008; Baragi et al., 2015; Wang et al., 2016) or increase growth rate and production only during the post-bloom phase of an experiment (Grossart et al., 2006; Sperling et al., 2013; Westwood et al., submitted). Thus, bacterial communities appear to be relatively tolerant to

ocean acidification, with bacterial growth indirectly affected by ocean acidification responses of the phytoplankton community (Grossart et al., 2006; Allgaier et al., 2008; Engel et al., 2013; Piontek et al., 2013; Sperling et al., 2013; Bergen et al., 2016).

    Mesocosm experiments are an effective way of monitoring the community response of microbial assemblages to environmental changes. Experiments examining multiple species and trophic levels can provide responses that differ significantly from mono-specific studies. Numerous mesocosm studies have now been performed, assessing the effect of ocean acidification on

natural marine microbial communities around the world (e.g. Kim et al., 2006; Hopkinson et al., 2010; Riebesell et al., 2013;



Paul et al., 2015; Bach et al., 2016; Bunse et al., 2016). Studies in the Arctic reported increases in phytoplankton primary productivity, growth, and organic matter concentration at $CO_2$ levels $\geq$800 $\mu atm$ under nutrient-replete conditions (Bellerby et al., 2008; Egge et al., 2009; Engel et al., 2013; Schulz et al., 2013), whilst the bacterial community was unaffected (Grossart et al., 2006; Allgaier et al., 2008; Paulino et al., 2008; Baragi et al., 2015). These studies also highlight the importance of

nutrient availability on the community response to elevated $CO_2$, with substantial differences in primary and bacterial productivity, Chlorophyll $a$ (Chl $a$), and elemental stoichiometry observed between nutrient-replete and nutrient-limited conditions (Riebesell et al., 2013; Schulz et al., 2013; Sperling et al., 2013; Bach et al., 2016).

Previous community-level studies investigating the effects of ocean acidification on natural coastal marine microbial communities in East Antarctica reported declines in primary and bacterial productivity when $CO_2$ levels exceeded 780 $\mu atm$

(Westwood et al., submitted). To build upon the results of Westwood et al. (submitted), a similar experimental design was utilised, with a natural marine microbial community from the same region exposed to $CO_2$ levels ranging from 343 to 1641 $\mu atm$ in 650 l minicosms. The methods were refined in our study to include an acclimation period to the $CO_2$ treatment under low light. Rates of primary productivity, bacterial productivity, and the accumulation of particulate organic matter (POM) were examined to ascertain whether the threshold for tolerance to $CO_2$ was similar to that reported by Westwood et al. (submitted),

or if acclimation affected the community response to high $CO_2$. Photophysiological measurements were also undertaken to assess underlying mechanisms that caused shifts in phytoplankton community productivity.

## 2   Methods

### 2.1   Minicosm setup

Natural microbial assemblages were incubated in six 650 l polyurethane tanks (minicosms) housed in a temperature controlled

shipping container. All minicosms were acid washed with 10% vol:vol AR HCl, thoroughly rinsed with MilliQ water and given a final rinse with seawater from the sampling site before use. The minicosms were filled with seawater taken amongst decomposing fast ice in Prydz Bay, at Davis Station, Antarctica (68° 35' S 77° 58' E) on 19th November 2014. Water was transferred by helicopter in multiple collections using a 720 l Bambi Bucket to fill a 7000 l polypropylene holding tank. Seawater was gravity fed into the minicosm tanks through Teflon lined hosing fitted with an in-line 200 $\mu m$ Arkal filter to

exclude metazooplankton. All minicosms were filled simultaneously to ensure uniform distribution of microbes in all tanks.

The ambient water temperature at the time of sampling in Prydz Bay was -1.0 °C. Tanks were temperature controlled to an average temperature of 0.0 °C, with a maximum range of $\pm$ 0.5 °C, through cooling of the shipping container and warming with two 300W aquarium heaters (Fluval) that were connected to a temperature control program via Carel temperature controllers. The contents of each tank were gently mixed by a shielded high density polyethylene auger, rotating at 15 rpm, and each tank

was covered with a sealed acrylic lid.

Each tank was illuminated on a 19:5 hr light:dark cycle by two 150W HQI-TS (Osram) metal halide lamps. The light output was initially filtered by one quarter colour temperature (CT) blue filter, two 90% neutral density (ND) filters (Arri), and a light-scattering filter, to produce low intensity light to slow phytoplankton growth in the 5 day acclimation period. During this




time, the $CO_2$ in each tank was gradually raised to the target concentration (Fig. 1, see below). At the conclusion of the $CO_2$ acclimation period, the light intensity was increased for 24 hrs through replacement of the two 90% ND filters with one 60% ND filter. The final light intensity was achieved with one quarter CT blue filter and light-scattering filter, which proved to be saturating for photosynthesis (see below).

Unless otherwise specified, samples were taken for analysis on days 1, 3, and 5 during the $CO_2$ acclimation period and every 2 days from day 8 to 18.

## 2.2    Carbonate chemistry measurements and calculations

Samples for carbonate chemistry measurements were collected daily from each minicosm in 500 ml glass stoppered bottles (Schott Duran) following the guidelines of Dickson et al. (2007). Sub-samples for dissolved inorganic carbon (DIC, 50 ml
glass stoppered bottles) and pH on the total scale ($pH_T$, 100 ml glass stoppered bottles) measurements were gently pressure filtered (0.2 µm) with a peristaltic pump at a flow rate of $\sim$30 ml min$^{-1}$, similar to Bockmon and Dickson (2014).

DIC was measured by infra-red absorption on an Apollo SciTech AS-C3 analyzer equipped with a LICOR LI-7000 detector using triplicate 1.5 ml samples. The instrument was calibrated (and checked for linearity) within the expected DIC concentration range with five sodium carbonate standards (Merck Suprapur) that were dried for 2 hours at 230 °C and prepared
gravimetrically in milliQ water (18.2 MΩ cm$^{-1}$) at 25 °C. Furthermore, daily measurements of certified reference material batch CRM127 (Dickson, 2010) were used for improved accuracy. Volumetrically measured DIC was converted to µmol kg$^{-1}$ using calculated density derived from known temperature and salinity. The typical precision among triplicate measurements was <2 µmol kg$^{-1}$.

$pH_T$ was measured spectrophotometrically (GBC UV-Vis 916) in a ten centimetre thermostated (25 °C) cuvette using the
pH indicator dye m-cresol purple (Acros Organics, 62625-31-4, Lot A0321770) following the approach described in Dickson et al. (2007), which included changes in sample pH due to dye addition. Contact with air was minimized by sample delivery, dye addition, and mixing via a syringe pump (Tecan, Cavro XLP6000). Dye impurities and instrument performance were accounted for by applying a constant off-set (+0.003 pH units), determined by the comparison of measured and calculated $pH_T$ (from known DIC and TA, including silicate and phosphate) of CRM127. Typical measurement precision for triplicates was
0.001 for higher and 0.003 for lower pH treatments. For further details see Schulz et al. (2017).

Carbonate chemistry speciation was calculated from measured DIC and $pH_T$. In a first step, at in situ measured salinities (WTW197 conductivity meter), practical alkalinity (PA) was calculated at 25 °C using the dissociation constants for carbonic acid determined by Mehrbach et al. (1973) as refitted by Lueker et al. (2000). Then, total carbonate chemistry speciation was calculated from measured DIC and calculated PA for in situ temperature conditions.

## 2.3    Carbonate chemistry manipulation

The fugacity of carbon dioxide ($f$CO$_2$) in the minicosms was adjusted by additions of $CO_2$ enriched natural seawater. In order to keep $f$CO$_2$ as constant as possible throughout the experiment, pH in each minicosm was measured with a portable, NBS-calibrated probe (Mettler Toledo) in the morning before sampling and the afternoon, to estimate the necessary amount of DIC





to be added. The required volume of $CO_2$ enriched seawater was then transferred into 1000 ml infusion bags and added to the individual minicosms at a rate of about $50 \, \mathrm{ml \, min^{-1}}$. After reaching target levels, the mean $CO_2$ levels in the minicosms were 343, 506, 634, 953, 1140, and 1641 µatm (Table S1).

## 2.4 Light irradiance

5 Average light intensity in each minicosm tank was calculated by measuring light intensity in the empty tanks at three depths (top, middle, and near-bottom) and across each tank (left, middle, and right) using a Biospherical Instruments' Laboratory Quantum Scalar Irradiance Meter (QSL-101). The average light irradiance received by phytoplankton within each tank was calculated following the equation of Riley (1957) (Table 1). Incoming irradiance ($\bar{I}_o$) was calculated as the average light intensity across the top of the tank. The average vertical light attenuation ($K_d$) was calculated as the slope from regression of 10 the natural log of light intensity at all three depths, and mixed depth ($Z_m$) was the depth of the minicosm tanks (1.14 m).

Changes in vertical light attenuation due to increases in Chl $a$ concentration throughout the experimental period were calculated from the equation in Westwood et al. (submitted); $K_{d(biomass)} = 0.0451157 \times$ Chl $a$ (mg m$^{-3}$). Total light attenuation $K_{d(total)}$ in each tank at each sampling day was calculated by addition of $K_d$ and $K_{d(biomass)}$.

## 2.5 Nutrient analysis

15 No nutrients were added to the minicosms during the experiment. Macronutrient samples were obtained from each minicosm following the protocol of Davidson et al. (2016). Seawater was filtered through 0.45 µm Sartorius filters into 50 ml Falcon tubes and frozen at -20 °C for analysis in Australia. Concentrations of nitrate plus nitrite (NOx), soluble reactive phosphorus (SRP), and molybdate reactive silica (Silica) were determined using flow injection analysis by Analytical Services Tasmania following Davidson et al. (2016).

## 20 2.6 Elemental analysis

Samples for POM analysis, particulate organic carbon (POC) and particulate organic nitrogen (PON), were collected following the method of Pearce et al. (2007). Equipment for sample preparation was soaked in Decon 20 (Decon Laboratories) for >2 days and thoroughly rinsed in MilliQ water before use. Forceps and cutting blades were rinsed in 100% acetone between samples. Seawater was filtered through muffled 25 mm Sartorius Quartz microfibre filters until clogged. The filters were folded 25 in half and frozen at -20 °C for analysis in Australia. Filters were thawed and opposite one-eighth sub-samples were cut and transferred into a silver POC cup (Elemental Analysis Ltd). Inorganic carbon was removed from each sample through addition of 20 µl of 2N HCl to each cup and drying at 60 °C for 36 hrs. When dry, each cup was folded shut, compressed into a pellet, and stored in desiccant until analysed at the Central Science Laboratory, University of Tasmania, using a Thermo Finnigan EA 1112 Series Flash Elemental Analyser.





## 2.7 Chlorophyll *a*

Seawater was collected from each minicosm and a measured volume was filtered through 13 mm Whatman GF/F filters for 20 mins. Filters were folded in half, blotted dry, and immediately frozen in liquid nitrogen for analysis in Australia. Pigments were extracted, analysed by HPLC, and quantified following the methods of Wright et al. (2010). Pigments (including Chl *a*)

were extracted from filters with 300 µl dimethylformamide plus 50 µl methanol, containing 140 ng apo-8'-carotenal (Fluka) internal standard, followed by bead beating and centrifugation to separate the extract from particulate matter. Extracts (125 µl) were diluted to 80% with water and analysed on a Waters HPLC using a Waters Symmetry C8 column and a Waters 996 photodiode array detector. Pigments were identified by comparing retention times and spectra to a mixed standard sample from known cultures (Jeffrey and Wright, 1997), run daily before samples. Peak integrations were performed using Waters Empower

software, checked manually for corrections, and quantified using the internal standard method (Mantoura and Repeta, 1997).

## 2.8 $^{14}$C primary productivity

Primary productivity incubations were performed following the method of Westwood et al. (2010), based on the technique of Lewis and Smith (1983). For all samples, 5.92 MBq (0.16 mCi) of $^{14}$C-sodium bicarbonate (NaH$^{14}$CO$_3$, PerkinElmer) was added to 162 ml of seawater from each minicosm, creating a working solution of 37 kBq ml$^{-1}$. Aliquots of this working

solution (7 ml) were then added to glass scintillation vials and incubated for 1 hr at 21 light intensities, ranging from 0 - 1412 µmol photons m$^{-2}$ s$^{-1}$. The temperature within each of the vials was maintained at -1.0 ± 0.3 °C through water cooling of the incubation chamber. The reaction was terminated with the addition of 250 µl of 6N HCl and the vials were shaken for 3 hours at 200 rpm to remove dissolved inorganic carbon. Duplicate time zero (T$_0$) samples were set up in a similar manner to determine background radiation, with 250 µl of 6N HCl added immediately to quench the reaction without exposure to light.

Duplicate 100% samples were also performed to determine the activity of the working solution for each tank. For each 100% sample, 100 µl of working solution was added to 7 ml 0.1M NaOH in filtered seawater to bind all $^{14}$C. For radioactive counts, 10 ml Ultima Gold LLT scintillation cocktail (PerkinElmer) was added to each scintillation vial, shaken, and decays per minute (DPM) were counted in a PerkinElmer Tri-Carb 2910TR Low Activity Liquid Scintillation Analyzer with a maximum counting time set at 3 min.

DPM counts were converted into primary productivity following the equation of Steemann Nielsen (1952) (Table 1), using measured DIC concentrations (varying between ∼2075- 2400 µmol kg$^{-1}$) and normalised to per unit Chl *a* using minicosm Chl *a* concentration (see above). Photosynthesis versus irradiance (PE) curves were modelled for each treatment following the equation of Platt et al. (1980) using the *Phytotools* package in R (Silsbe and Malkin, 2015; R Core Team, 2016). Photosynthetic parameter estimates included the light-saturated photosynthetic rate (P$_{max}$), maximum photosynthetic efficiency (α),

photoinhibition rate (β), and saturating irradiance (E$_k$). Modelled Chl *a*-specific primary productivity (csPP$_{14C}$) was calculated following the equation of Platt et al. (1980) using average minicosm light irradiance ($\bar{I}$). Gross primary production rates (GPP$_{14C}$) in each tank were calculated from modelled csPP$_{14C}$ and Chl *a* concentration (see above). Calculations and units for each parameter are presented in Table 1.



## 2.9 Gross community productivity

Community photosynthesis and respiration rates were measured using custom-made mini-chambers. The system consisted of four, 5.1 ml glass vials with oxygen sensor spots (Pyroscience) attached on the inside of the vials using non-toxic silicon glue. The vials were sealed, ensuring any oxygen bubbles were omitted and all vials were stirred continuously using small Teflon magnetic fleas to allow homogenous mixing of gases within the system during measurements. To improve signal to noise ratio, seawater from each minicosm was concentrated above a 0.8 μm, 47 mm diameter polycarbonate membrane filter (Poretics) with gentle vacuum filtration and re-suspended in seawater from each minicosm $CO_2$ treatment. Each chamber was filled with the cell suspension and placed in a temperature controlled incubator ($0.0 \pm 0.5\,°C$). Light was supplied via fluorescent bulbs above each chamber and light intensity calibrated using a $4\pi$ sensor. Oxygen optode spots were connected to a FireSting $O_2$ logger and data acquired using FireSting software (PyroScience). The optode was calibrated according to the manufacturer's protocol immediately prior to measurements using a freshly prepared sodium thiosulfate solution (10% w/w) and agitated filtered seawater (0.2 μm) at experimental temperature for 0% and 100% air saturation values, respectively. Oxygen concentration was recorded until a linear change in rate was established for each pseudoreplicate (n = 4). Measurements were recorded in the dark and subsequently in the light (188 μmol photons $m^{-2}\,s^{-1}$). Chl $a$ concentration in each sample was determined through spectrophotometric analysis at the University of Technology, Sydney, using a 90% chilled acetone extraction procedure. Gross primary productivity ($GPP_{O_2}$) was then calculated from dark respiration ($Resp_{O_2}$) and net primary production ($NPP_{O_2}$) rates and normalised to Chl $a$ concentration (Table 1).

## 2.10 Chlorophyll $a$ fluorescence

Photosynthetic efficiency of the microalgal community was measured via Chl $a$ fluorescence using a Pulse Amplitude Modulated fluorometer (Water PAM, Walz). A 3 ml aliquot from each minicosm was transferred into a quartz cuvette with continuous stirring to prevent cells from settling. To establish an appropriate dark adaptation period, several replicates were measured after 5, 10, 15, 20 and 30 min of dark adaptation, with the latter having the highest maximum quantum yield of PSII ($F_v/F_m$). Following dark adaptation, minimum fluorescence ($F_0$) was recorded before application of a high intensity saturating pulse of light (saturating pulse width = 0.8 s; saturating pulse intensity >3,000 μmol photons $m^{-2}\,s^{-1}$), where maximum fluorescence ($F_m$) was determined. From these two parameters the maximum quantum yield of PSII was calculated (Schreiber, 2004). Following $F_v/F_m$, a five-step steady state light curve (SSLC) was conducted with each light level (130, 307, 600, 973, 1450 μmol photons $m^{-2}\,s^{-1}$) applied for 5 min before recording the light-adapted minimum ($F_t$) and maximum fluorescence ($F_{m'}$) values. Each light step was spaced by a 30 sec dark 'recovery' period, before the next light level was applied. Three pseudoreplicate measurements were conducted on each minicosm sample at 0.1 °C. Non-photochemical quenching (NPQ) of Chl $a$ fluorescence was calculated from $F_m$ and $F_{m'}$ measurements. Relative electron transport rates (rETR) were calculated as the product of effective quantum yield ($\Delta F/F_{m'}$) and actinic irradiance ($I_a$). Calculations and units for each parameter are presented in Table 1.





### 2.11 Community carbon concentrating mechanism activity

To investigate the effects of $CO_2$ on carbon uptake, two inhibitors for carbonic anhydrase (CA) were applied to the 343 and 1641 µatm treatments on day 15: ethoxzolamide (EZA, Sigma), which inhibits both intracellular carbonic anhydrase (iCA) and extracellular carbonic anhydrase (eCA), and acetazolamide (AZA, Sigma), which blocks eCA only. Stock solutions of EZA (20 mM) and AZA (5 mM) were prepared in milliQ water, and the pH adjusted using NaOH to minimise pH changes when added to the samples. Before fluorometric measurements were made, water samples from the 343 and 1641 µatm $CO_2$ treatments were filtered into $\geq$10 and <10 µm fractions and aliquots were inoculated either with 50 µl of milliQ water adjusted with NaOH (control) or 50 µM final concentration of chemical inhibitor (EZA and AZA). Fluorescence measurements of size fractionated control- and inhibitor-exposed cells were performed using the Water-PAM. A 3 ml aliquot of sample was transferred into a quartz cuvette, with stirring, and left in the dark for 30 min before maximum quantum yield of PSII ($F_v/F_m$) was determined (as described above). Actinic light was then applied at 1450 µmol photons m$^{-2}$ s$^{-1}$ for 5 min before effective quantum yield of PSII ($\Delta F/F_{m'}$) was recorded. Three pseudoreplicate measurements were conducted on each minicosm sample at 0.1 °C.

### 2.12 Bacterial abundance

Bacterial abundance was determined daily using a Becton Dickinson FACScan or FACSCalibur flow cytometer fitted with a 488 nm laser following the protocol of Thomson et al. (2016). Samples were pre-filtered through a 50 µm mesh (Nitex), stored at 4 °C in the dark, and analysed within 6 hr of collection. Samples were stained for 20 min with 1:10,000 dilution SYBR-Green I (Invitrogen) (Marie et al., 2005) and PeakFlow Green 2.5 µm beads (Invitrogen) were added to the sample as an internal fluorescence standard. Three pseudoreplicate samples were prepared from each minicosm seawater sample. Samples were run for 3 mins at a low flow rate ($\sim$12 µl min$^{-1}$) and bacterial abundance was determined from side scatter (SSC) versus green (FL1) fluorescence bivariate scatter plots. The analysed volume was calibrated to the sample run time and each sample was run for precisely 3 min, resulting in an analysed volume of 0.0491 and 0.02604 ml on the FACSCalibur and FACScan, respectively. The volume analysed was then used to calculate final cell concentrations.

### 2.13 Bacterial productivity

Bacterial productivity measurements were performed following the Leucine incorporation by microcentrifuge method of Kirchman (2001). Briefly, 70 nM $^{14}$C-Leucine (PerkinElmer) was added to 1.7 ml seawater from each minicosm in 2 ml polyethylene Eppendorf tubes and incubated for 2 hr in the dark at 4 °C. Three pseudoreplicate samples were prepared from each minicosm seawater sample. The reaction was terminated by the addition of 90 µl 100% trichloroacteic acid (TCA, Sigma) to each tube. Duplicate background controls were also performed following the same method, with 100% TCA added immediately before incubation. After incubation, samples were spun for 15 min at 12,500 rpm and the supernatant was removed. The cell pellet was resuspended into 1.7 ml ice-cold 5% TCA and spun again for 15 min at 12,500 rpm and the supernatant removed. The cell pellet was then resuspended into 1.7 ml ice-cold 80% ethanol, spun for a further 15 min at 12,500 rpm and the supernatant removed. The cell pellet was allowed to dry completely before addition of 1 ml Ultima Gold scintillation cocktail (PerkinElmer).



The Eppendorf tubes were placed into glass scintillation vials and DPMs were counted in a PerkinElmer Tri-Carb 2910TR Low Activity Liquid Scintillation Analyzer with a maximum counting time of 3 min.

DPM counts were converted to $^{14}$C-Leucine incorporation rates following the equation in Kirchman (2001) and used to calculate gross bacterial production (GBP$_{14C}$), following Simon and Azam (1989). Bacterial production was divided by total

bacterial abundance to determine cell-specific bacterial productivity within each treatment (csBP$_{14C}$). Calculations and units for each parameter are presented in Table 1.

### 2.14 Statistical analysis

The minicosm experimental design measured the microbial community growth in six unreplicated $f$CO$_2$ treatments. Therefore, sub-samples from each minicosm were within-treatment pseudoreplicates and thus, only provide a measure of the variability of

the within-treatment sampling procedure. We use pseudoreplicates as true replicates in order to provide an informal assessment of differences among treatments, noting that results must be treated as indicative and interpreted conservatively.

A linear or quadratic regression model was fitted to each CO$_2$ treatment over time using the *Stats* package in R (R Core Team, 2016) and an omnibus test of differences between treatments was assessed by ANOVA. This analysis ignored the repeated measures nature of this data, which could not be modelled due to the low number of time points and an absence of

replication at each time. For the CCM activity measurements, differences between treatments were tested by one-way ANOVA, followed by a post-hoc Tukey's test to determine which treatments differed. The significance level for all tests was set at $<$ 0.05.

## 3  Results

### 3.1  Carbonate chemistry

The $f$CO$_2$ of each treatment was modified in a step-wise fashion over 5 days to allow for acclimation of the microbial community to the changed conditions. Target treatment conditions were reached in all tanks by day 5, ranging from 343 to 1641 μatm, equating to an average pH$_T$ of 8.1 to 7.45 (Fig. 1, Table S1). The initial seawater was calculated to have an $f$CO$_2$ of 356 μatm and a PA of 2317 μmol kg$^{-1}$, from a measured pH$_T$ of 8.08 and DIC of 2187 μmol kg$^{-1}$ (Fig. S1, Table S2). One minicosm was maintained close to these conditions (343 μatm) throughout the experiment as a control treatment.

### 25  3.2  Light climate

The average light irradiance for all CO$_2$ treatments is presented in Table S3. During the CO$_2$ acclimation period (days 1-5) the average light irradiance was $0.88 \pm 0.2$ μmol photons m$^{-2}$ s$^{-1}$ and was increased to an average light irradiance of 86.6 $\pm$ 20.5 μmol photons m$^{-2}$ s$^{-1}$ by day 8. The average vertical light attenuation (K$_d$) across all minicosm tanks was $0.92 \pm$ 0.2. Increased Chl $a$ concentration over time in all CO$_2$ treatments increased K$_{d(total)}$ from $0.96 \pm 0.01$ on day 1 to $2.6 \pm$





0.3 on day 18. This resulted in a decline in average light irradiance between days 8-18 from $86.61 \pm 20.5$ to $35.97 \pm 9.3$ µmol photons m$^{-2}$ s$^{-1}$.

### 3.3 Nutrients

Nutrient concentrations were similar across all treatments at the beginning of the experiment (Table S2) and did not change during the acclimation period (days 1-5). NOx fell from $26.2 \pm 0.74$ µM on day 8 to concentrations below detection limits on day 16 in the 343 (control), 634, and 953 µatm treatments (Fig. 2a). In the remaining treatments (506, 1140, and 1641 µatm), NOx fell below detection limits on day 18, with the slowest draw-down in the 1641 µatm treatment. SRP concentrations were initially $1.74 \pm 0.02$ µM and all CO$_2$ treatments followed a similar draw-down sequence to NOx, reaching very low concentrations (0.13 µM) on day 18 in all treatments (Fig. 2b). In contrast, silica was replete in all treatments throughout the experiment falling from $60.0 \pm 0.91$ µM to $43.6 \pm 2.45$ µM (Fig. 2c). Draw-down of silica began an exponential decline from day 8 and followed a similar pattern in treatments to NOx and SRP, with the most silica draw-down in the 634 µatm treatment and the least in the 1641 µatm treatment.

### 3.4 Particulate organic matter

Particulate organic carbon (POC) and nitrogen (PON) concentrations were initially low, $4.7 \pm 0.15$ and $0.5 \pm 0.98$ µM respectively, and increased after day 8 in all treatments (Fig 3a-b). The accumulation of POC and PON was effectively the reciprocal of the draw-down of nutrients (see above), being lowest in the high CO$_2$ treatments ($\geq 1140$ µatm) and highest in the 343 and 643 µatm treatments. Rates of POC and PON accumulation were both affected by nutrient exhaustion, with the largest declines in the 343 and 634 µatm treatments between days 16 to 18. POC and PON concentrations on day 18 were highest in the 953 µatm treatment. The ratio of POC to PON (C:N) was similar for all treatments, declining from $8.0 \pm 0.38$ on day 8 to $5.7 \pm 0.28$ on day 16 (Fig, 3c). The slowest initial decline in C:N ratio occurred in the 1641 µatm treatment, displaying a prolonged lag until day 10, after which it decreased to values similar to all other treatments. Nutrient exhaustion on day 18 coincided with an increase in the C:N ratio in all treatments, with C:N ratios >10 in the 343, 634, and 953 µatm treatments and lower C:N ratios in the 506, 1140, and 1641 µatm treatments (8.6-6.7).

### 3.5 Chlorophyll *a*

Chl *a* concentration was low at the beginning of the experiment, $0.91 \pm 0.16$ µg l$^{-1}$ and increased in all treatments after day 8 (Fig 4a). Chl *a* accumulation rates were similar amongst treatments $\leq 634$ µatm up until day 14, with a slight promotion in Chl *a* concentration in the 506 and 634 µatm treatments on day 16 compared to the control treatment. By day 18, only the 503 µatm treatment remained higher than the control. Chl *a* accumulation rates in the 953 and 1140 µatm treatments were initially slow but increased after day 14, with Chl *a* concentrations similar to the control on days 16-18. The highest CO$_2$ treatment (1641 µatm) had the slowest rates of Chl *a* accumulation, displaying a lag in growth between days 8-12, after which Chl *a* concentration increased but remained lower than the control. Rates of Chl *a* accumulation slowed between days 16 to 18 in all





treatments except 1641 µatm, coinciding with nutrient limitation. At day 18, the highest Chl $a$ concentration was in the 506 µatm exposed treatment and lowest at 1641 µatm. The omnibus test of an interaction between day and tank was significant ($F_{5,23}$ = 5.45, p = 0.002; Table S4), indicating at least one treatment was appreciably different from the control treatment. Examination of individual coefficients from the model revealed that only the highest $CO_2$ treatment group, 1641 µatm, was

judged significantly different from the control at the 5% level.

### 3.6    $^{14}$C primary productivity

During the $CO_2$ and light acclimation phase of the experiment (days 1-8) all treatments displayed a steady decline in the maximum photosynthetic rate ($P_{max}$) and the maximum photosynthetic efficiency ($\alpha$) to levels on day 8 approximately half that at the beginning of the experiment, suggesting cellular acclimation to the light conditions (Fig. S2a-b). Thereafter, relative

to the control, $P_{max}$ and $\alpha$ were lowest in $CO_2$ levels $\geq$953 µatm and $\geq$634 µatm, respectively. Rates of photoinhibition ($\beta$) and saturating irradiance ($E_k$) were variable and did not differ among treatments (Fig. S2c-d). The average $E_k$ across all treatments was 28.7 $\pm$ 8.6 µmol photons m$^{-2}$ s$^{-1}$, indicating that the light intensity in the minicosms was saturating for photosynthesis (see above) and not photoinhibitory ($\beta$ < 0.002 µg C (µg Chl $a$)$^{-1}$ (µmol photons m$^{-2}$ s$^{-1}$)$^{-1}$ h$^{-1}$).

Chl $a$-specific primary productivity (csPP$_{14C}$) and gross primary production (GPP$_{14C}$) was low during the $CO_2$ acclimation

(days 1-5) and increased with increasing light climate after day 5. Rates of csPP$_{14C}$ in treatments $\geq$634 µatm $CO_2$ were consistently lower than the control between days 8-16, with the lowest rates in the highest $CO_2$ treatment (1641 µatm) (Fig. S3a). Rates of GPP$_{14C}$ in treatments $\leq$953 were similar between days 8-16, with the 343 (control), 506, and 953 µatm treatments increasing to 46.7 $\pm$ 0.34 µg C l$^{-1}$ h$^{-1}$ by day 18 (Fig. 4b). Compared to these treatments, GPP$_{14C}$ in the 634 µatm treatment was lower on day 18, only reaching 39.7 µg C l$^{-1}$ h$^{-1}$, possibly due to the concurrent limitation of NOx in this treatment on

day 16 (see above).

The omnibus test of an interaction between day and tank was significant ($F_{5,23}$ = 4.95, p = 0.003; Table S5), indicating at least one treatment was appreciably different from the control treatment. Examination of the significance of individual curve terms revealed this manifested as differences between the 1140 and 1641 µatm treatments and the control group at the 5% level. No other curves were different from the control. In particular, GPP$_{14C}$ in the 1641 µatm treatment was very low until day 12, after

which it increased steadily until day 16. Between days 16-18, a substantial increase in GPP$_{14C}$ was observed in this treatment, subsequently resulting in a rate on day 18 that was similar to the 1140 µatm treatment (36.3 $\pm$ 0.08 µg C l$^{-1}$ h$^{-1}$). Although, these treatments never reached rates of GPP$_{14C}$ as high as the control.

### 3.7    Gross community productivity

Productivity of the phytoplankton community increased over time in all $CO_2$ treatments, however there were clear differ-

ences in the timing and magnitude of this increase between treatments (Fig. 4c). A $CO_2$ effect was evident on day 12, where Chl $a$-normalised gross $O_2$ productivity rates (GPP$_{O_2}$) increased with increasing $CO_2$ level, ranging from 0.7-7.8 µmol $O_2$ (µg Chl $a$)$^{-1}$ h$^{-1}$. After day 12, the communities in $CO_2$ treatments $\leq$634 µatm continued to increase their rates of GPP$_{O_2}$ through to day 18 (3.3 $\pm$ 0.6 µmol $O_2$ (µg Chl $a$)$^{-1}$ h$^{-1}$). The 953 and 1140 µatm $CO_2$ treatments peaked on day 12



(3.7 and 4.4 µmol $O_2$ (µg Chl $a$)$^{-1}$ h$^{-1}$, respectively) and then declined on day 14 to rates similar to the control treatment. In contrast, the 1641 µatm treatment maintained high rates of GPP$_{O_2}$ from days 12-14 (8.2 ± 0.6 µmol $O_2$ (µg Chl $a$)$^{-1}$ h$^{-1}$), coinciding with the recovery of photosynthetic health (F$_v$/F$_m$, see below) and the initiation of growth in this treatment (see above). After this time, rates of GPP$_{O_2}$ declined in this treatment to rates similar to the control.

## 3.8 Community photosynthetic efficiency

Community maximum quantum yield of PSII (F$_v$/F$_m$) showed a dynamic response over the duration of the experiment (Fig. 5). Values initially increased during the low light $CO_2$ adjustment period, but declined by day 8 once irradiance levels were increased. Between days 8-14, differences were evident in the community photosynthetic health across the $CO_2$ treatments, although by day 16 these differences had disappeared. Steady state light curves revealed that the community photosynthetic response did not change with increasing $CO_2$. Effective quantum yield of PSII ($\Delta$F/F$_{m'}$) and NPQ showed no variability with $CO_2$ treatment (Fig. S4-S5). There was however, a notable decline in overall NPQ in all tanks with time, indicating an adjustment to the higher light conditions. Relative electron transport rates (rETR) showed differentiation with respect to $CO_2$ at high light (1450 µmol photons m$^{-2}$ s$^{-1}$) on days 10-12. However, as seen with the F$_v$/F$_m$ response, this difference was diminished by day 18 (Fig. S6).

## 3.9 Community CCM activity

There was a significant decline in the effective quantum yield of PSII ($\Delta$F/F$_{m'}$) with the addition of the iCA inhibitor EZA to both the large (≥10 µm, p = 0.02) and small (<10 µm, p < 0.001) size fractions of the phytoplankton community exposed to the control (343 µatm) $CO_2$ treatment (Fig. 6a). The addition of EZA to cells under high $CO_2$ (1641 µatm) had no effect on $\Delta$F/F$_{m'}$ for either size fraction. However, in the case of the small cells under high $CO_2$ (Fig. 6b), $\Delta$F/F$_{m'}$ was the same as that measured in the control $CO_2$ in the presence of EZA. The addition of AZA, which inhibits eCA, had no effect for either $CO_2$ treatments in the large celled community. In contrast, there was a significant decline in $\Delta$F/F$_{m'}$ in the smaller fraction in the control $CO_2$ treatment (p < 0.001), but no effect of AZA addition under high $CO_2$. Again, the high $CO_2$ cells exhibited the same $\Delta$F/F$_{m'}$ as those measured under the control $CO_2$ in the presence of AZA.

## 3.10 Bacterial abundance

During the 8 day acclimation period, bacterial abundance in treatments ≥634 µatm increased with increasing $CO_2$, reaching 26.0-32.4 x 10$^7$ cells l$^{-1}$, and remaining high until day 13 (Fig. 7a). Between days 7-13, bacterial abundances in $CO_2$ treatments ≥953 were higher then the control. In contrast, abundance remained constant in treatments ≤506 µatm (20.6 ± 1.4 x 10$^7$ cells l$^{-1}$) through to day 11. Cell numbers rapidly declined in all treatments after day 12, finally stabilising at 0.5 ± 0.2 x 10$^7$ cells l$^{-1}$. An omnibus test of differences between treatments revealed a significant interaction between day and tank ($F_{5,185}$ = 9.78, p < 0.001; Table S6), indicating at least one $CO_2$ treatment was different from the control treatment. Examina-




tion of individual coefficients from the model revealed that $CO_2$ treatments $\geq 953$ μatm were significantly different from the control at the 5% level.

### 3.11 Bacterial productivity

Gross bacterial production ($GBP_{14C}$) was low all $CO_2$ treatments ($0.2 \pm 0.03$ μg C l$^{-1}$ h$^{-1}$) and changed little during the first 5 days of incubation (Fig. 7b). Thereafter it increased, coinciding with exponential growth in the phytoplankton community. The most rapid increase in $GBP_{14C}$ was observed in the 634 μatm treatment, resulting in a rate twice that of all other treatments by day 18 (2.1 μg C l$^{-1}$ h$^{-1}$). No difference was observed between all other treatments, increasing to an average rate of $1.1 \pm 0.1$ μg C l$^{-1}$ h$^{-1}$ by day 18. Cell-specific bacterial productivity ($csBP_{14C}$) was low in all treatments ($1.2 \pm 0.5$ fg C l$^{-1}$ h$^{-1}$) until day 14, with slower rates in treatments $\geq 953$ μatm, likely due to high cell abundances observed in these treatments (Fig. S3b). It then increased from day 14, coinciding with a decline in bacterial abundance. Rates of $csBP_{14C}$ rates did not differ among treatments up until day 18, when the 634 μatm treatment was higher than all other treatments (0.5 pg C cell$^{-1}$ l$^{-1}$ h$^{-1}$).

### 4 Discussion

Our study of a natural Antarctic phytoplankton community identified a critical threshold for tolerance of $CO_2$ between 953 and 1140 μatm, above which photosynthetic health was negatively affected and rates of carbon fixation and Chl $a$ accumulation declined. Low rates of primary productivity also led to declines in nutrient uptake rates and POM production, although there was no effect of $CO_2$ on C:N ratios, indicating that ocean acidification effects on the phytoplankton community did not modify organic matter stoichiometry. In contrast, bacterial productivity was unaffected by increased $CO_2$. Instead, their production coincided with increased organic matter supply from phytoplankton primary productivity. In the following sections these effects will be investigated further, with suggestions for the possible mechanisms that may be driving the responses observed.

### 4.1 Ocean acidification effects on phytoplankton productivity

The results of this study suggest that exposing phytoplankton to high $CO_2$ levels can decouple the two stages of photosynthesis. At $CO_2$ levels $\geq 1140$ μatm, $GPP_{O_2}$ increased strongly yet displayed the lowest rates of carbon fixation ($GPP_{14C}$). Photosynthetic fixation of carbon is a two stage process (Reviewed in Behrenfeld et al., 2004). In the first stage, light-dependent reactions occur within the chloroplast, converting light energy (photons) into the cellular energy products, adenosine triphosphate (ATP) and nicotinamide adenine dinucleotide phosphate (NADPH), producing $O_2$ as a by-product. This cellular energy is then utilised in a second, light-independent pathway, which uses the carbon-fixing enzyme RuBisCo to convert $CO_2$ into sugars through the Calvin Cycle. However, under certain circumstances the relative pool of energy may also be consumed in alternative pathways, such as respiration and photoprotection (Behrenfeld et al., 2004; Gao and Campbell, 2014). Increases in energy requirements for these alternate pathways have been demonstrated, where measurements of maximum photosynthetic rates ($P_{max}$) and photosynthetic efficiency ($\alpha$) display changes that result in no change to saturating irradiance levels ($E_k$) (Behrenfeld et al., 2004,



2008; Halsey et al., 2010). This "$E_k$-independent variability" was evident in our study, where decreases in $P_{max}$ and $\alpha$ were observed in the high $CO_2$ treatments, while $E_k$ remained unaffected.

This highlights an important tipping point in the phytoplankton community's ability to cope with the energetic requirements for maintaining efficient productivity under high $CO_2$. While studies on individual phytoplankton species have reported decou-
pling of the photosynthetic pathway under conditions of stress, to date, no studies on natural phytoplankton communities have reported this response. Under laboratory conditions, stresses such as nutrient limitation (Halsey et al., 2010) or a combination of high $CO_2$ and light climate (Hoppe et al., 2015; Liu et al., 2017) have been shown to induce such a response, where isolated phytoplankton species possess higher energy requirements for carbon fixation. In our study, the phytoplankton community experienced a dynamic light climate due to continuous gentle mixing of the minicosm contents, and although nutrients weren't
limiting, the phytoplankton in the higher $CO_2$ treatments did show lower csPP$_{14C}$ rates, which could be linked to higher energy demand for light-independent processes. Since nutrients were replete and not a likely source of stress, it follows that $CO_2$ and light were likely the only sources of stress on this community.

Increased respiration rates could account for the decreased carbon fixation rates measured. Thus far, respiration rates are commonly reported as either unaffected or lower under increasing $CO_2$ (Hennon et al., 2014; Trimborn et al., 2014; Spilling
et al., 2016). This effect is generally attributed to declines in cellular energy requirements, via processes such as down-regulation of CCMs, which can result in observed increases to rates of production (Spilling et al., 2016). Despite this, decreased growth rates have been linked to enhanced respiratory carbon loss at high $CO_2$ levels (800-1000 µatm) (Gao et al., 2012b). Community respiration rates in our study increased with increasing $CO_2$ (data not shown). However, they were generally proportional to the increase in $O_2$ production (i.e, the ratio of production to respiration remained constant across $CO_2$ conditions),
making it unlikely to be a significant contributor to the decline in carbon fixation.

Another process known to consume energy when extracellular pH is lowered, is the activation of proton pumps to maintain cellular homeostasis (Taylor et al., 2012). Under normal conditions, when the extracellular environment is above pH 7.8, excess $H^+$ ions generated within the cell are able to passively diffuse out of the cell. However, a lowering of the oceanic pH below 7.8 is likely to halt the passive removal of internal $H^+$, requiring the utilisation of energy-intensive proton pumps (Taylor et al.,
2012), and thus potentially reducing the energy pool available for carbon fixation. Our results are consistent with this idea of a critical pH threshold, as the declines in GPP$_{14C}$ were observed in treatments $\geq 1140$ µatm, the $CO_2$ treatments where the pH ranged from 7.69-7.45.

Despite the initial stress of high $CO_2$ between days 8-12, the phytoplankton community displayed a strong ability to adapt to these conditions. The $CO_2$-induced reduction in $F_v/F_m$ showed a steady recovery between days 12 and 16, with all treatments
displaying similar high $F_v/F_m$ at day 16 (0.68-0.71). This recovery in photosynthetic health suggests that the phytoplankton community was able to acclimate to the high $CO_2$ conditions, possibly through cellular acclimation, changes in community structure, or most likely, a combination of both. Cellular acclimations were observed in our study. A lowering of NPQ and a minimisation of the $CO_2$-related response to photoinhibition (rETR) at high light intensity suggested that PSII was being down-regulated to adjust to a higher light climate. Decreased energy requirements for carbon fixation were also observed in
the photosynthetic pathway, resulting in increases to GPP$_{14C}$ and Chl *a* accumulation rates. Acclimation to increased $CO_2$ has





been reported in a number of studies, resulting in shifts in carbon and energy utilisation (Sobrino et al., 2008; Hopkinson et al., 2010; Hennon et al., 2014; Trimborn et al., 2014; Zheng et al., 2015). Numerous photophysiological investigations on individual phytoplankton species also report species-specific tolerances to increased $CO_2$ (Gao et al., 2012a; Gao and Campbell, 2014; Trimborn et al., 2013, 2014) and a general trend toward smaller-celled communities with increased $CO_2$ has been reported in

ocean acidification studies globally (Schulz et al., 2017). Changes in community structure were observed with increasing $CO_2$, with taxon-specific thresholds of $CO_2$ tolerance (Hancock et al., submitted). Within the diatom community, the response was also related to size, leading to an increase in abundance of small ($<20\ \mu m$) diatoms in the high $CO_2$ treatments ($\geq 953\ \mu atm$). Therefore, the community acclimation observed is likely driven by an increase in growth of more tolerant species.

It is often suggested that the down-regulation of CCMs help to moderate the sensitivity of phytoplankton communities to

increasing $CO_2$. The carbon-fixing enzyme RubisCO has a low affinity for $CO_2$ that is compensated for through CCMs that actively increase the intracellular $CO_2$ to saturating concentrations (Raven, 1991; Badger, 1994; Badger et al., 1998; Hopkinson et al., 2011). This process requires additional cellular energy (Raven, 1991) and numerous studies have suggested that the energy savings from down-regulation of CCMs in phytoplankton could explain increases in rates of primary productivity at elevated $CO_2$ levels (Tortell et al., 2000; Tortell and Morel, 2002; Cassar et al., 2004; Tortell et al., 2008b, 2010; Trimborn et al.,

2013; Young et al., 2015). Although, Young et al. (2015) showed that the energetic costs of CCMs in Antarctic phytoplankton are low and thus, any down-regulation of CCMs at increased $CO_2$ would provide little benefit. We show that the CCM carbonic anhydrase (CA) was utilised by the phytoplankton community at our control $CO_2$ level (343 $\mu atm$) and was down-regulated at high $CO_2$ (1641 $\mu atm$). No promotion of primary productivity was observed, supporting the notion that whilst this may alleviate energy supply constraints, it does not lead to increased rates of carbon fixation (Rost et al., 2003; Cassar et al., 2004;

Riebesell, 2004).

Size-specific differences in phytoplankton CCM utilisation were also observed at control $CO_2$. The absence of eCA activity in the large cells ($\geq 10\ \mu m$) suggests that bicarbonate ($HCO_3^-$) was the dominant carbon source used by this fraction of the phytoplankton community (Burkhardt et al., 2001; Tortell et al., 2008a). This is not surprising as direct $HCO_3^-$ uptake has been commonly reported among Antarctic phytoplankton communities (Cassar et al., 2004; Tortell et al., 2008a, 2010).

Down-regulation of iCA in large cells with high $CO_2$ suggests that passive diffusion of $CO_2$ into the cells increased, lowering the requirement for intracellular conversion of $HCO_3^-$ to $CO_2$. As both direct $HCO_3^-$ uptake and iCA activity are energetic processes (Raven, 1991) this may increase energy efficiency within the cell (Tortell et al., 2008b). However, the decline in growth of the phytoplankton community at high $CO_2$ suggests that these energy savings were utilised elsewhere. On the other hand, the small cell community ($<10\ \mu m$) used both iCA and eCA at control $CO_2$, implying carbon was sourced for

photosynthesis through both extracellular conversion of $HCO_3^-$ to $CO_2$ and direct $HCO_3^-$ uptake (Rost et al., 2003). The prymnesiophyte, *Phaeocystis antarctica* dominated this size fraction in all treatments (Hancock et al., submitted) and has been found to have a strong preference for $CO_2$ as a carbon source (Trimborn et al., 2013). Interestingly, a significant decline in photosynthetic health was also observed in the small cell fraction at 1641 $\mu atm$, coinciding with a strong decline in *P. antarctica* cell abundance at this $CO_2$ level (Hancock et al., submitted). Thus, it is likely that the small cells reflect the response of *P.*

*antarctica* to increased $CO_2$.





The presence of iCA has also been proposed as a possible mechanism for increased sensitivity of phytoplankton to decreased pH conditions. Satoh et al. (2001) found that the presence of iCA caused strong intracellular acidification and inhibition of carbon fixation when a $CO_2$-tolerant iCA-expressing algal species was transferred from ambient conditions to very high $CO_2$ (40%). Down-regulation of iCA through acclimation in a 5% $CO_2$ treatment eliminated this response, with similar tolerance observed in an algal species with low ambient iCA activity. Thus, the down-regulation of iCA activity at high $CO_2$, as was seen in this study, may not only decrease cellular energy demands but may also be operating as a cellular protection mechanism, allowing the cell to maintain intracellular homeostasis.

Contrary to high the $CO_2$ treatments, the phytoplankton community appeared to tolerate $CO_2$ levels up to 953 µatm, identifying a $CO_2$ threshold. Between days 8-14 we observed a small and insignificant $CO_2$-related decline in $F_v/F_m$, $GPP_{14C}$, and Chl $a$ accumulation between the 343-953 µatm treatments. Tolerance of $CO_2$ levels up to ∼1000 µatm has often been observed in natural phytoplankton communities in regions exposed to fluctuating $CO_2$ levels. In these communities increasing $CO_2$ often has no effect on primary productivity (Tortell et al., 2000; Tortell and Morel, 2002; Tortell et al., 2008b; Hopkinson et al., 2010; Tanaka et al., 2013; Sommer et al., 2015; Young et al., 2015; Spilling et al., 2016) or growth (Tortell et al., 2008b; Schulz et al., 2013), although an increase in primary production has been observed in some instances (Riebesell, 2004; Tortell et al., 2008b; Egge et al., 2009; Tortell et al., 2010; Hoppe et al., 2013; Holding et al., 2015). Previous studies in Prydz Bay report a tolerance of the phytoplankton community to $CO_2$ levels up to 750 µatm (Davidson et al., 2016; Thomson et al., 2016; Westwood et al., submitted). The most likely reason for this high tolerance is that these communities are already exposed to highly variable $CO_2$ conditions. $CO_2$ naturally builds beneath the fast ice in winter, when primary productivity is low (Perrin et al., 1987; Legendre et al., 1992), and is rapidly depleted during spring and summer by phytoplankton blooms, resulting in annual $fCO_2$ fluctuations between 50-433 µatm (Gibson and Trull, 1999; Roden et al., 2013). Thus, variable $CO_2$ environments appear to promote adaptations within the phytoplankton community to manage the stress imposed by fluctuating $CO_2$.

Changes in POM production and C:N ratio in phytoplankton communities can have significant effects on carbon sequestration and change their nutritional value for higher trophic levels (Finkel et al., 2010; van de Waal et al., 2010; Polimene et al., 2016). We observed a decline in particulate organic matter production (POM) at $CO_2$ levels ≥1140 µatm, while organic matter stoichiometry (C:N ratio) appeared to be predominantly controlled by nutrient consumption. Increases in POM production were similar to Chl $a$ accumulation, with declines in high $CO_2$ treatments (≥1140µatm) due to low rates of primary productivity. Carbon overconsumption has been reported in some natural phytoplankton communities exposed to increased $CO_2$, resulting in increases in the C:N ratio (Riebesell et al., 2007; Engel et al., 2014). While in our study the C:N ratio did decline to below the Redfield Ratio during exponential growth, it remained within previously reported C:N ratios of coastal phytoplankton communities in this region (Gibson and Trull, 1999; Pasquer et al., 2010). Therefore, it is difficult to say whether or not changes in primary productivity will affect organic matter stoichiometry in this region, particularly as any resultant long-term changes in community composition may have a future influence (Finkel et al., 2010).





### 4.2 Ocean acidification effects on bacterial productivity

In contrast to the phytoplankton community, bacteria were tolerant of high $CO_2$ levels. The low bacterial productivity and abundance of the initial community is characteristic of the post-winter bacterial community in Prydz Bay where their growth is limited by organic nutrient availability (Pearce et al., 2007). Whilst an increase in cell abundance was observed at $CO_2$ levels $\geq 634$ µatm, it was possible that this response was driven by a decline in grazing by heterotrophs (Thomson et al., 2016; Westwood et al., submitted) instead of a direct $CO_2$-related promotion in bacterial growth. The subsequent decline in abundance was likely due to top-down control from the heterotrophic nanoflagellate community, which displayed an increase in abundance at this time (Hancock et al., submitted). Bacterial tolerance to high $CO_2$ has been reported previously in this region (Thomson et al., 2016; Westwood et al., submitted) and has also been reported in numerous studies in the Arctic (Grossart et al., 2006; Allgaier et al., 2008; Paulino et al., 2008; Baragi et al., 2015; Wang et al., 2016), suggesting that the marine bacterial community will be resilient to increasing $CO_2$.

While we detected an increase in bacterial productivity, this response appeared to be correlated with an increase in Chl *a* concentration and available POM, rather than $CO_2$. Bacterial productivity was similar among all $CO_2$ treatments, except for a final promotion of productivity at 634 µatm on day 18. This promotion of growth may be linked to an increase in diatom abundance observed in this treatment (Hancock et al., submitted). The coupling of bacterial growth with phytoplankton productivity has been reported by numerous studies on natural marine microbial communities (Allgaier et al., 2008; Grossart et al., 2006; Engel et al., 2013; Piontek et al., 2013; Sperling et al., 2013; Bergen et al., 2016). Thus, it is likely that the bacterial community was controlled more by grazing and nutrient availability than by $CO_2$ level.

### 5 Conclusions

These results support the identification of a tipping point in the marine microbial community response to $CO_2$ between 953 and 1140 µatm. When exposed to $CO_2$ beyond their limits of tolerance, declines in growth rates, primary productivity, and organic matter production were observed in the phytoplankton community. Despite this, the community displayed the ability to adapt to these high $CO_2$ conditions, down-regulating CCMs and likely adjusting other intracellular mechanisms to cope with the added stress of low pH. However, the lag in growth and subsequent acclimation to high $CO_2$ conditions allowed for more tolerant species to thrive (Hancock et al., submitted).

Conditions in Antarctic coastal regions fluctuate throughout the seasons and the marine microbial community is already tolerant to changes in $CO_2$ level, light availability, and nutrients (Gibson and Trull, 1999; Roden et al., 2013). It is possible that phytoplankton communities already exposed to highly variable conditions will be more capable of adapting to the projected changes in $CO_2$ (Schaum and Collins, 2014). Although, this will likely also include adaptation at the community level, causing a shift in dominance to more tolerant species. This has been observed in numerous ocean acidification experiments, with a trend in community composition favouring picophytoplankton and away from large diatoms (Davidson et al., 2016; Reviewed in Schulz et al., 2017). Such a change in phytoplankton community composition may have flow on effects to higher trophic levels that feed on Antarctic phytoplankton blooms. It could also have a significant effect on the biological pump, with decreased




carbon draw-down at high $CO_2$, causing a negative feedback on anthropogenic $CO_2$ uptake. Coincident increases in bacterial abundance under high $CO_2$ conditions may also increase the efficiency of the microbial loop, resulting in increased organic matter remineralisation and further declines in carbon sequestration.

*Data availability.* Experimental data used for analysis are available via the Australian Antarctic Data Centre (DOI being created).

5  *Author contributions.* AD, KW, and KP concieved and designed the experiments. AD led and oversaw the minicosm experiment. SD and KP performed the experiments and data analysis. KS performed the carbonate system measurements and manipulation. IP performed pigment extraction and analysis. JM provided statistical guidance. SD wrote the manuscript with significant input from KP, KS, and AD. All authors provided contributions and critical review of the manuscript.

*Competing interests.* The authors declare that they have no conflict of interest.

10  *Acknowledgements.* This study was funded by the Australian Government, Department of Environment and Energy as part of Australian Antarctic Science Project 4026 at the Australian Antarctic Division and an Elite Research Scholarship awarded by the Institute for Marine and Antarctic Studies, University of Tasmania. We would like to thank Prof. Andrew McMinn for valuable comments on our manuscript, Penelope Pascoe for the flow cytometric analyses, Cristin Sheehan for photosynthetis and respiration data, and Dr. Thomas Rodemann from the Central Science Laboratory, University of Tasmania for elemental analysis of our POM samples. We gratefully acknowledge the 15  assistance of AAD technical support in designing and equipping the minicosms and Davis Station expeditioners in the summer of 2014/15 for their support and assistance.



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





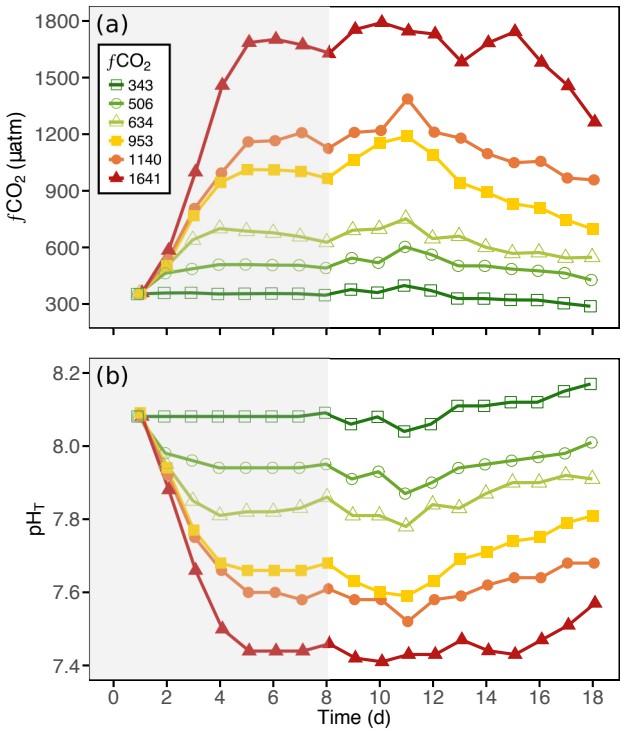

**Figure 1.** (a) $f\mathrm{CO_2}$ and (b) $\mathrm{pH}_T$ conditions within each of the minicosm treatments over time. Grey shading indicates $\mathrm{CO_2}$ and light acclimation period.





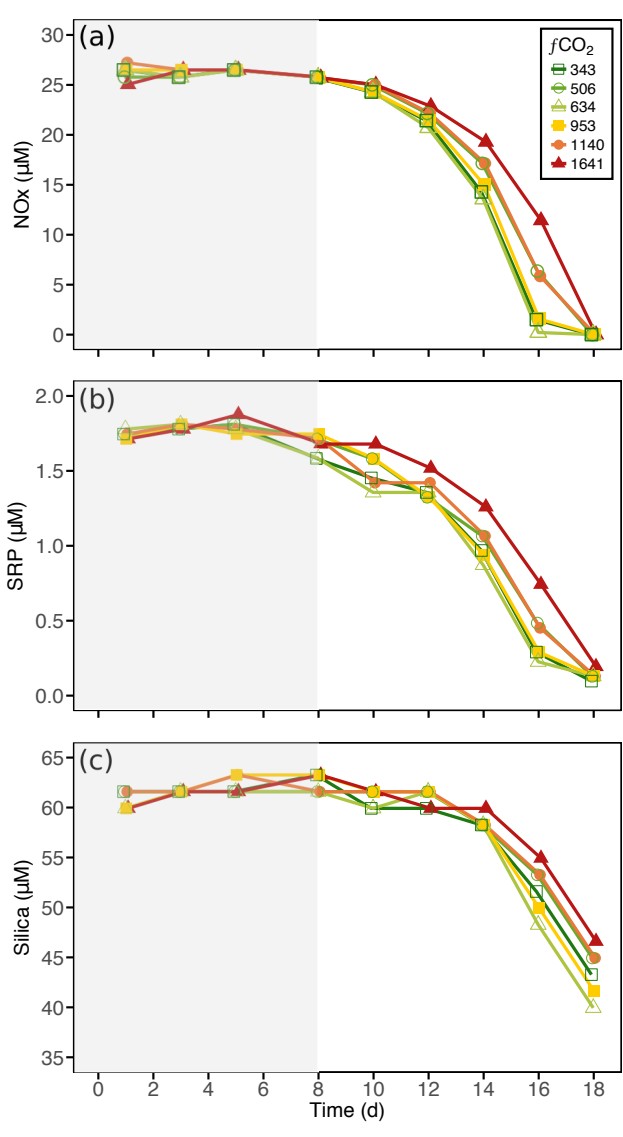

**Figure 2.** Nutrient draw-down within each of the minicosm treatments over time. (a) Nitrate + nitrite (NOx), (b) soluble reactive phosphorus (SRP), and (c) molybdate reactive silica (Silica). Grey shading indicates $CO_2$ and light acclimation period.



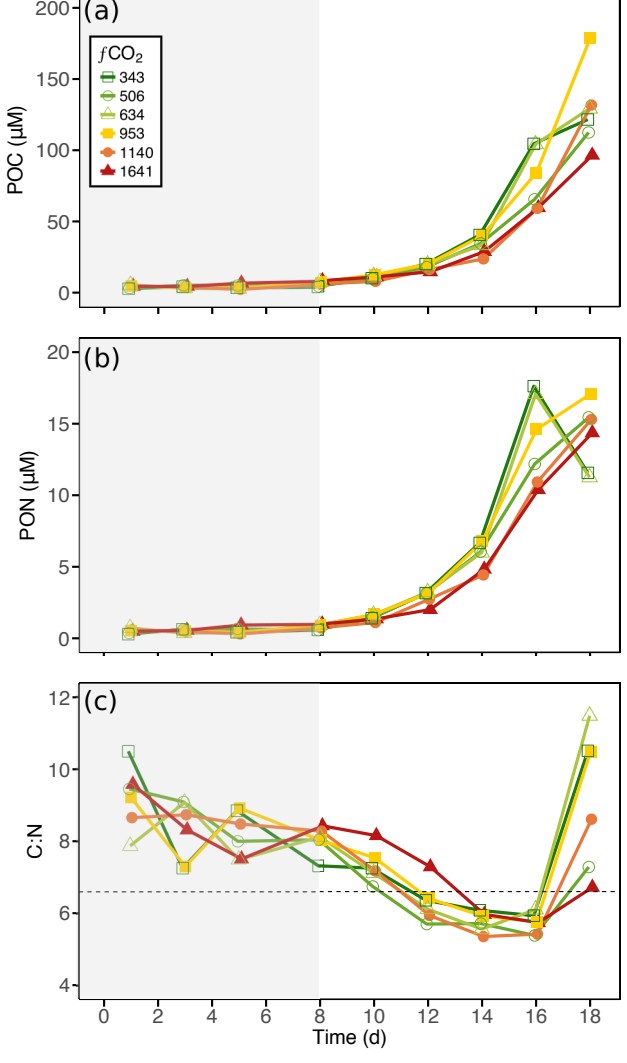

**Figure 3.** Particulate organic matter concentration and C:N ratio of each of the minicosm treatments over time. (a) Particulate organic carbon (POC), (b) particulate organic nitrogen (PON), and (c) carbon:nitrogen (C:N) ratio. Dashed line indicates C:N Redfield Ratio of 6.6. Grey shading indicates $CO_2$ and light acclimation period.



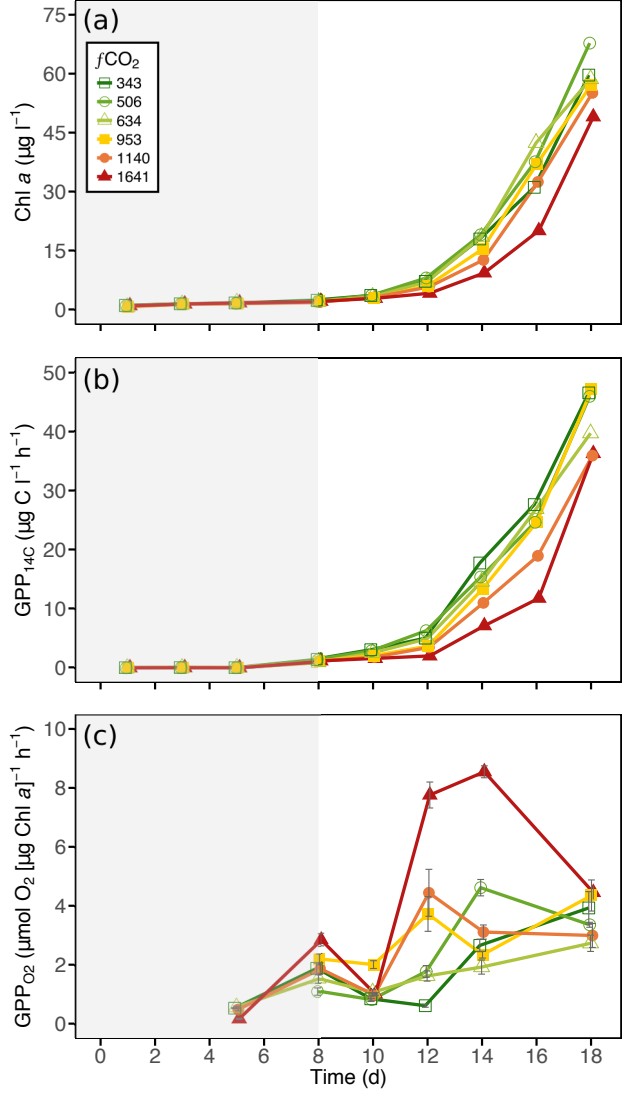

**Figure 4.** Phytoplankton biomass accumulation and community primary production in each of the minicosm treatments over time. (a) Chlorophyll $a$ (Chl $a$) concentration, (b) $^{14}$C-derived gross primary production (GPP$_{14C}$), and (c) $O_2$-derived gross primary productivity (GPP$_{O_2}$). Error bars display one standard deviation of pseudoreplicate samples. Grey shading indicates $CO_2$ and light acclimation period.





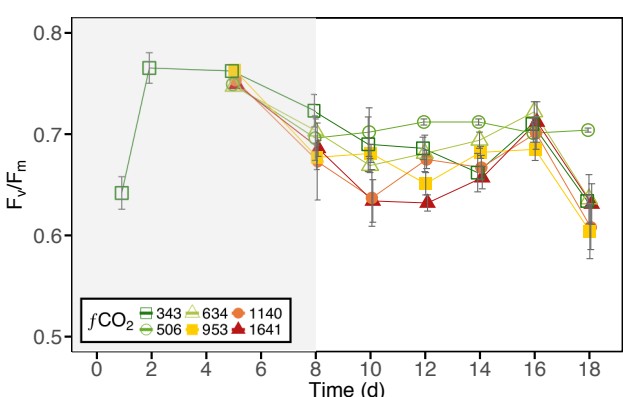

**Figure 5.** Maximum quantum yield of PSII ($F_v/F_m$) in each of the minicosm treatments over time. Error bars display one standard deviation of pseudoreplicate samples. Grey shading indicates $CO_2$ and light acclimation period.



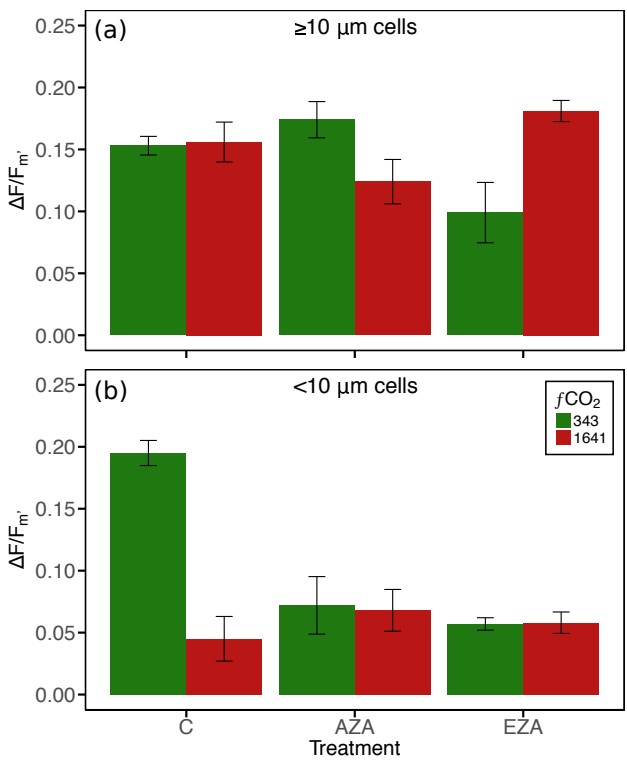

**Figure 6.** Effective quantum yield of PSII ($\Delta F/F_{m'}$) of (a) large ($\geq$10 µm) cells and (b) small ($<$10 µm) cells in the control (343 µatm) and high (1641 µatm) $CO_2$ treatments treated with carbonic anhydrase (CA) inhibitors. A decline in $\Delta F/F_{m'}$ with application of inhibitor indicates CCM activity. C: control, no CA inhibitor, AZA: acetazolamide, blocks extracellular carbonic anhydrase, EZA: ethoxzolamide, blocks intracellular and extracellular carbonic anhydrase. Error bars display one standard deviation of pseudoreplicate samples.





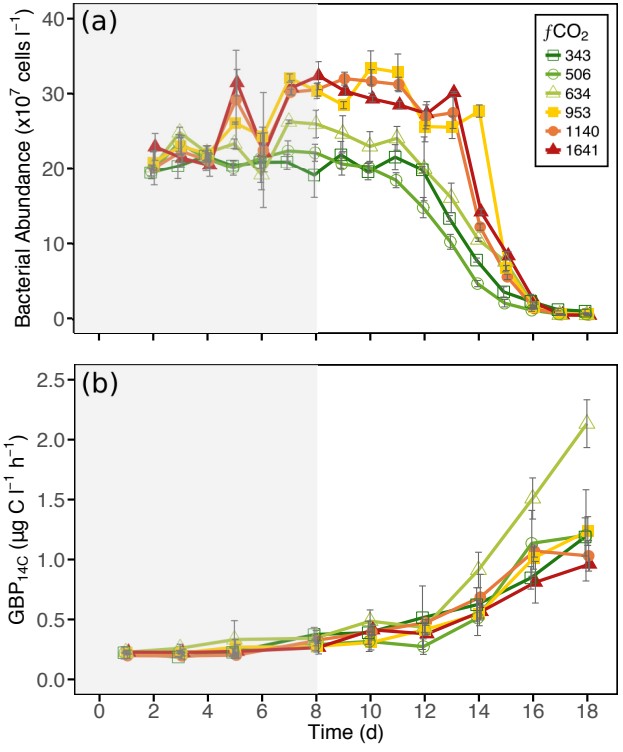

**Figure 7.** Bacterial abundance and community production in each of the minicosm treatments over time. (a) Bacterial cell abundance and (b) [14]C-derived gross bacterial production (GBP$_{14C}$). Error bars display one standard deviation of pseudoreplicate samples. Grey shading indicates $CO_2$ and light acclimation period.

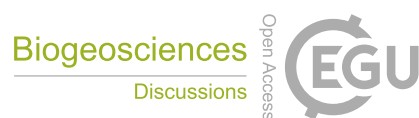



**Table 1.** Definitions, measurements and calculations for productivity data

| Name | Definition | Units | Measurements and calculations |
|---|---|---|---|
| *Primary Productivity* | | | |
| Carbon incorporation | Total $^{14}$C-sodium bicarbonate incorporation | µg C (µg Chl $a$)$^{-1}$ l$^{-1}$ h$^{-1}$ | Equation from Steemann Nielsen (1952) $= \dfrac{(DPM_s - DPM_{T_0})}{DPM_{100\%}} \times DIC \times 1.05 \text{ / time / Chl } a$ |
| $\alpha$ | Maximum photosynthetic efficiency | µg C (µg Chl $a$)$^{-1}$ (µmol photons m$^{-2}$ s$^{-1}$)$^{-1}$ h$^{-1}$ | Modelled from PE curve of 21 light intensities, 0 - 1411 µmol photons m$^{-2}$ s$^{-1}$ |
| $\beta$ | Photoinhibition rate | µg C (µg Chl $a$)$^{-1}$ (µmol photons m$^{-2}$ s$^{-1}$)$^{-1}$ h$^{-1}$ | Modelled from PE curve of 21 light intensities, 0 - 1411 µmol photons m$^{-2}$ s$^{-1}$ |
| $P_{max}$ | Maximum photosynthetic rate | µg C (µg Chl $a$)$^{-1}$ l$^{-1}$ h$^{-1}$ | Equation from Platt et al. (1980) $= P_s \times \dfrac{\alpha}{(\alpha + \beta)} \times \dfrac{\beta}{(\alpha + \beta)}^{\frac{\beta}{\alpha}}$ |
| $E_k$ | Saturating irradiance | µmol photons m$^{-2}$ s$^{-1}$ | Equation from Platt et al. (1980) $= \dfrac{P_{max}}{\alpha}$ |
| $\bar{I}$ | Average irradiance received by phytoplankton cells | µmol photons m$^{-2}$ s$^{-1}$ | Equation from Riley (1957) $= \bar{I}_o(1 - e^{(-K_d \times Z_m)})/(K_d \times Z_m)$ |
| csPP$_{14C}$ | $^{14}$C Chl $a$-specific primary productivity | µg C (µg Chl $a$)$^{-1}$ h$^{-1}$ | Equation from Platt et al. (1980) $= P_s \times e^{\frac{-\alpha \bar{I}}{P_s}} \times e^{\frac{-\beta \bar{I}}{P_s}}$ |
| GPP$_{14C}$ | $^{14}$C gross primary production | µg C l$^{-1}$ h$^{-1}$ | $= $ csPP$_{14C} \times$ Chl $a$ |
| GPP$_{O_2}$ | O$_2$ gross primary productivity | µmol O$_2$ (µg Chl $a$)$^{-1}$ h$^{-1}$ | $= $ NPP$_{O_2}$ + Resp$_{O_2}$ / Chl $a$ |
| *Photophysiology* | | | |
| F$_v$/F$_m$ | Maximum quantum yield of PSII | | $= \dfrac{F_m - F_O}{F_m}$ |
| ΔF/F$_{m'}$ | Effective quantum yield of PSII | | $= \dfrac{F_{m'} - F}{F_{m'}}$ |
| rETR | Relative electron transport rate | | $= \dfrac{\Delta F_v}{F_{m'}} \times I_a$ |
| NPQ | Non-photochemical quenching | | $= \dfrac{F_m - F_{m'}}{F_{m'}}$ |
| *Bacterial Productivity* | | | |
| nmol Leucine$_{inc}$ | Moles of exogenous $^{14}$C-Leucine incorporated | nmol l$^{-1}$ h$^{-1}$ | Equation from Kirchman (2001) $= (DPM_s - DPM_{t_0})$ / time / 2.22 x10$^6$ × SA (nmol µCi$^{-1}$) / sample vol (l) |
| GBP$_{14C}$ | $^{14}$C gross bacterial production | µg C l$^{-1}$ h$^{-1}$ | Equation from Simon and Azam (1989) $= $ (nmol Leucine$_{inc}$ / 10$^3$) × 131.2 / 0.073 ×0.86 × 2 |
| csBP$_{14C}$ | $^{14}$C cell-specific bacterial productivity | fg C cell$^{-1}$ l$^{-1}$ h$^{-1}$ | $= $ GBP$_{14C}$ / cells l$^{-1}$ |

$P_s$: maximum photosynthetic output with no photoinhibition, from Platt et al. (1980), DPM: sample DPM, SA: specific activity of $^{14}$C-Leucine isotope
All other abbreviations defined in Methods Section