# Peer review of "Ocean acidification of a coastal Antarctic marine microbial community reveals a critical threshold for CO2 tolerance in phytoplankton productivity"

_Biogeosciences, 2017_

## Referee Comment (RC1) · Anonymous Referee #1 · 1 Aug 2017

General comments: In this work, Deppeler et al. installed six minicosm to study how ocean acidification will affect coastal microbial communities, including photoautotrophs and heterotrophs. This kind of field work is rather difficult to conduct, because it requires large amount of resources, participation of different groups and limited by meteorological condition and logistical support. They stated that there existed a tipping point for CO2 effects, ocean acidification with CO2>1140 uatm would decrease primary production of phytoplankton, while no consistent effects on bacteria. This is an interesting finding, however, the data analysis is inadequate, especially for the threshold,

the author should present a fig, the x-axis is pCO2, and y-axis could be GPP, FV/FM or other parameters, to clearly show there is a tipping point. Overall, this manuscript is well structured, while there are some flaws need to be fixed. Specific comments: Introduction: This section is well written, reflected the background of this study Method: I recommend the author to present a picture of whole scene of the minicosm, that will be much easier for the reader to follow the method. P4 Line24, the seawater was transferred from another location by helicopter, my impression is that the community structure might be different with the local seawater where the experiment done. The major concern is that seawater in minicosm might contact with local seawater during the manipulation, is the contamination even for all minicosms? Because you don't have replication for each CO2, even the contamination happened differently for minicosm, while the statistics cannot tell you. P4 Line 33 Why you use blue filter? Are the transmission spectra available? P6 Line 17, Why ammonium was not measured? It is actually an important nutrient for phytoplankton. P9 Line 5, Are AZ and EZ directly dissolved in milliQ water? I remember these two reagents are quite difficult to dissolve in pure water. Here is just a reminder. P10 Line 7, I understand that it is impossible to run 6 CO2 with replicates, however, I think the author should do more job on statistics instead of simple comparison with ANOVA. They could try to do some curve fitting, e.g. exponential rising for POC, PON, Chla, decay of nutrients etc, to extract some valuable numbers for comparison. Results: This section is well written Discussion: This section is somewhat redundant, the author talked too much about CCM. CCMs are quite complicated and involved by many proteins, enzymes, and ion channels. The present data is obtained only using two CA inhibitors, so to what extend these data can reflect the activity of CCM? Moreover, you only measured chlorophyll fluorescence, which is direct measurement of light reaction, however, CA only participates in CO2 acquisition for dark reaction, so the measured parameters further limit the interpretation of data for CCM. I suggest the author to compress CCM related paragraph. P14 Line 22 "photosynthetic . . . process", this is a very short sentence, please rephrased.

---

## Referee Comment (RC2) · Anonymous Referee #2 · 10 Aug 2017

This manuscript uses 6 minicosms to investigate the effect of CO2 on the Antarctica microbial plankton (phytoplankton and bacteria) community. The authors' conclude there is a critical threshold for CO2 and above this threshold of 953 -1140 uatm, phytoplankton productivity diminishes, with no observable effect on bacterial production. The great advantage of minicosms is their capability to test a community response, however, they are large undertakings, requiring significant investment of time, resources and people and thus results are often split into multiple publications, as is the case here. Unfortunately, without the data in the other manuscripts, we get only a partial

story and it is difficult to give an accurate review.

Overall the authors' did a good job on the CO2 manipulations and the manuscript is well written. While there have already been a number of minicosm experiments with CO2 manipulations, most polar studies have focused on the Arctic and it is interesting to see an Antarctic focus on this scale. As a general comment, minicosm experiments often produce conflicting results and there should be more effort discussing possible mechanisms that underlie the variable results between experiments. For example, the authors' mention how their results differ from other studies but do not provide possible explanations of why e.g. differing setup, differing communities etc.

My main concern for this manuscript is that I am not convinced the results support their conclusion of a CO2 threshold between 953 and 1140 uatm. Only GPP14C showed that treatments over 953uatm CO2 had lower productivity. In other figures, either only 1641 uatm appeared different, no significant difference was found, or a mid range CO2 treatment was an outlier. The only statistical analysis they used was ANOVA, which identifies statistically different treatments instead of looking for trends related to CO2 concentrations. Because of the type of statistical test chosen, only a threshold rather than a CO2 trend was tested.

I understand that other results are being published in other papers but considering there are no replicates, the authors need to do a better job reassuring the readers that the differences between minicosms are directly a response to CO2 and not due to other changes e.g. community shifts. The methods section details how community composition was measured but no results were presented and instead will be presented in Hancock et al.

The authors' focuses their story around CO2 with little mention of the effects of pH. I think this should be expanded upon.

There are a few issues with the 14C and O2 measurements used for GPP. There have been a number of studies demonstrating that incubating for 1h for 14C does not capture GPP, and that O2 respiration in the dark does not always equal respiration in the light. Both would result in errors in GPP. While this data can be used (as it is hard to measure true GPP), these caveats should be acknowledged in the manuscript. The units used for GPP based on 14C and GPP based on O2 are different, making them difficult to compare. Comparison would provide an idea on whether there is a realistic photosynthetic quotient, and this would also go a long way as to helping interpret NPQ and other non-carbon assimilatory processes.

The authors' state that CO2 had no effect on bacterial production. However, looking at figure 7, there appears to be higher bacterial abundance between days 8 – 14 in the high CO2 treatments, which are not observed in bacterial productivity, indicating that bacterial production per cell is lower at high CO2? Surely, this is a CO2 response?

The C:N data for POM is interesting but it is hard to discount carbon overconsumption without also looking at DOM. This would also be useful in interpreting the GPP 14C results. Respiration rates would also be useful. I realize these measurements can't be taken but the authors' should discuss these factors.

In the methods section the authors' should justify the length of acclimation, why it was done under low light and why a blue filter was used.

---

## Author Comment (AC1) · 22 Sep 2017

Response to Reviewer 1

General comments: In this work, Deppeler et al. installed six minicosm to study how ocean acidification will affect coastal microbial communities, including photoautotrophs and heterotrophs. This kind of field work is rather difficult to conduct, because it requires large amount of resources, participation of different groups and limited by meteorological condition and logistical support. They stated that there existed a tipping

point for CO2 effects, ocean acidification with CO2>1140 uatm would decrease primary production of phytoplankton, while no consistent effects on bacteria. This is an interesting finding, however, the data analysis is inadequate, especially for the threshold, the author should present a fig, the x-axis is pCO2, and y-axis could be GPP, FV/FM or other parameters, to clearly show there is a tipping point. Overall, this manuscript is well structured, while there are some flaws need to be fixed.

Response: We present figures of production over time in each CO2 treatment and statistically compare these response surfaces to identify CO2-induced differences among treatments. To address the issue raised by the reviewer we drafted graphs showing the rate of productivity against fCO2 at each time with the intention of adding a threshold value (Fig. 1). We thank the reviewer for their suggestion as plotting our data versus fCO2 allowed us to better discriminate the trends in rates of Chl a accumulation and 14C-GPP among the CO2 treatments. This visualisation of the data showed that the downturn in these parameters occurred between 634 and 953 uatm fCO2 and could be discerned following $\geq$ 12 days incubation. In addition, we acknowledge that nutrient limitation confounds our ability to determine an fCO2 threshold on the final day of the experiment (day 18). We will include a figure in the manuscript and provide further consideration of these results in the text.

Specific comments: Introduction: This section is well written, reflected the background of this study Method: I recommend the author to present a picture of whole scene of the minicosm, that will be much easier for the reader to follow the method.

Response: We have added in a figure showing a photograph of several minicosm tanks to aid our description of the minicosm setup.

P4 Line24, the seawater was transferred from another location by helicopter, my impression is that the community structure might be different with the local seawater where the experiment done. The major concern is that seawater in minicosm might contact with local seawater during the manipulation, is the contamination even for all

minicosms? Because you don't have replication for each CO2, even the contamination happened differently for minicosm, while the statistics cannot tell you.

Response: We believe this comment is borne of a misinterpretation of our methods, which should be clarified by our inclusion of the photograph above. The only time local seawater (from immediately offshore) was added to the contents of the minicom tanks was in the daily additions of <1 l of 0.22 um filtered and CO2-saturated seawater that was added to the contents of each tank to return the fCO2 back to the target value. We had omitted stating that this seawater was 0.22 um filtered before CO2 enrichment and subsequent addition to the minicosms and have added this to the Methods.

P4 Line 33 Why you use blue filter? Are the transmission spectra available?

Response: A quarter CT blue filter was used to convert the tungsten lighting to a daylight spectral distribution. This is achieved by attenuating wavelengths <500 nm by ~20% and >550 nm by ~40%. The transmission spectra of the 150W HQI-TS/NDL metal halide lamps is available online at www.osram.com.au/media/resource/ hires/335357/powerstar-hqi-ts-excellence-70-w-and-150-w—the-latest-innovation-in-quartz-tec.pdf. We have included this information the manuscript.

P6 Line 17, Why ammonium was not measured? It is actually an important nutrient for phytoplankton.

Response: We agree that ammonium is an important nutrient for phytoplankton growth and we did measure the concentration of this nutrient in our tanks. However, we omitted showing this data because it rapidly fell below detection limits (by day 12) and showed no CO2 treatment-related differences. We have updated the manuscript to include ammonium and our justification for omitting it from analysis in our results.

P9 Line 5, Are AZ and EZ directly dissolved in milliQ water? I remember these two reagents are quite difficult to dissolve in pure water. Here is just a reminder.

[Figure]

Response: It is true that many studies use DMSO or Acetone to dissolve EZ and AZ, but there are also some papers that have used MilliQ water, (e.g. Young et al., 2001). We acknowledge the reviewer's comment, however, we encountered no problems with the solubility of these reagents in MilliQ water.

P10 Line 7, I understand that it is impossible to run 6 $CO_2$ with replicates, however, I think the author should do more job on statistics instead of simple comparison with ANOVA. They could try to do some curve fitting, e.g. exponential rising for POC, PON, Chla, decay of nutrients etc, to extract some valuable numbers for comparison.

Response: The statistical analysis did include curve fitting using quadratic regression models. The ANOVA analysis was performed to test for significant differences among these curves. We recognise that this may not have been clear in the manuscript and have updated our results and the table captions in the Supplementary file.

Results: This section is well written Discussion: This section is somewhat redundant, the author talked too much about CCM. CCMs are quite complicated and involved by many proteins, enzymes, and ion channels. The present data is obtained only using two CA inhibitors, so to what extend these data can reflect the activity of CCM? Moreover, you only measured chlorophyll fluorescence, which is direct measurement of light reaction, however, CA only participates in $CO_2$ acquisition for dark reaction, so the measured parameters further limit the interpretation of data for CCM. I suggest the author to compress CCM related paragraph.

Response: We are grateful for the reviewer's comment and have condensed the CCM discussion. We have also included a sentence to highlight the limitation to our interpretation, having only measured light reactions.

P14 Line 22 "photosynthetic . . . process", this is a very short sentence, please rephrased.

Response: We have reviewed this sentence and rephrased it as the reviewer has requested.

References:Young E, Beardall J, Giordano M (2001) Inorganic carbon acquisition by Dunaliella tertiolecta (Chlorophyta) involves external carbonic anhydrase and direct HCO3- utilization insensitive to the anion exchange inhibitor DIDS. European Journal of Phycology 36 (1):81-88

———————————————

[Figure]

[Figure]

**Fig. 1.** CO2 threshold analysis for chlorophyll a accumulation, 14C-gross primary productivity rate, and accumulation of particulate organic nitrogen

---

## Author Comment (AC2) · 22 Sep 2017

Response to Reviewer 2

This manuscript uses 6 minicosms to investigate the effect of CO2 on the Antarctica microbial plankton (phytoplankton and bacteria) community. The authors conclude there is a critical threshold for CO2 and above this threshold of 953 -1140 uatm, phytoplankton productivity diminishes, with no observable effect on bacterial production. The great advantage of minicosms is their capability to test a community response, however, they

are large undertakings, requiring significant investment of time, resources and people and thus results are often split into multiple publications, as is the case here. Unfortunately, without the data in the other manuscripts, we get only a partial story and it is difficult to give an accurate review.

Overall the authors' did a good job on the CO2 manipulations and the manuscript is well written. While there have already been a number of minicosm experiments with CO2 manipulations, most polar studies have focused on the Arctic and it is interesting to see an Antarctic focus on this scale. As a general comment, minicosm experiments often produce conflicting results and there should be more effort discussing possible mechanisms that underlie the variable results between experiments. For example, the authors' mention how their results differ from other studies but do not provide possible explanations of why e.g. differing setup, differing communities etc.

Response: We agree with the reviewer's observation and have added in further discussion comparing our findings with those in others studies.

My main concern for this manuscript is that I am not convinced the results support their conclusion of a CO2 threshold between 953 and 1140 uatm. Only GPP14C showed that treatments over 953uatm CO2 had lower productivity. In other figures, either only 1641 uatm appeared different, no significant difference was found, or a mid range CO2 treatment was an outlier. The only statistical analysis they used was ANOVA, which identifies statistically different treatments instead of looking for trends related to CO2 concentrations. Because of the type of statistical test chosen, only a threshold rather than a CO2 trend was tested.

Response: We thank the reviewer for their comments and have performed additional analysis of the data in order to highlight a threshold value (Fig. 1, see also response 1 to reviewer 1). While the statistical analysis indicated that significant differences among our treatments were few, plotting our data versus fCO2 allowed us to look at the CO2 trends and identify a downturn in Chl a accumulation and rates of 14C-GPP

at $CO_2$ concentrations between 634 and 953 uatm. We were also able to show that this downturn could be seen from day 12 but could not be ascribed on day 18 due to nutrient limitation in some $fCO_2$ treatments. We will include a figure in the manuscript and provided additional discussion of our results in this context. We would also like to specify that the ANOVA analysis was performed to statistically compare curves fitted to the Chl a and 14C-GPP values in each treatment over time. We recognise that this may not have been clear in the original manuscript and have updated our results and the table captions in the Supplementary file.

I understand that other results are being published in other papers but considering there are no replicates, the authors need to do a better job reassuring the readers that the differences between minicosms are directly a response to $CO_2$ and not due to other changes e.g. community shifts. The methods section details how community composition was measured but no results were presented and instead will be presented in Hancock et al.

Response: The response we saw could be elicited by effects of the physiology of individual cells in the community studied and/or changes in the community composition that favoured $CO_2$-tolerant taxa. The bulk photosynthetic parameters of the phytoplankton community we present cannot differentiate among these effects. We have provided a summary of the community compositional changes observed in the minicosms in the discussion (P15 Line 5-8). We have not provided an in-depth analysis of the community compositional changes because they have been submitted as a companion paper in the same special edition of Biogeosciences.

It is possible the reviewer refers to the community composition derived from the HPLC analysis of pigment composition in our study. This analysis was only used to quantify the Chl a concentration in this manuscript. The wording has been modified to make this clearer. The authors' focuses their story around $CO_2$ with little mention of the effects of pH. I think this should be expanded upon.

[Figure]

Response: We strongly agree that the change in pH does have an effect on the cellular physiology. We have expanded the discussion around the effect of pH on cellular homeostasis in the discussion and have included further references to studies that have investigated this in natural phytoplankton communities.

There are a few issues with the 14C and O2 measurements used for GPP. There have been a number of studies demonstrating that incubating for 1h for 14C does not capture GPP, and that O2 respiration in the dark does not always equal respiration in the light. Both would result in errors in GPP. While this data can be used (as it is hard to measure true GPP), these caveats should be acknowledged in the manuscript. The units used for GPP based on 14C and GPP based on O2 are different, making them difficult to compare. Comparison would provide an idea on whether there is a realistic photosynthetic quotient, and this would also go a long way as to helping interpret NPQ and other non-carbon assimilatory processes.

Response: The authors agree with the reviewer and thank them for drawing our attention to these points. In revising the data and the paper, we have identified a mistake in our methods section. The authors in fact measured post-illumination respiration rates so as to better estimate GPP-O2. Where the initial steeper part of the slope directly after illumination is used to try and better capture the true respiration rate in the light. We have amended our methods and results to clarify this point. We have also updated our methods section to acknowledge the caveats of 14C GPP measurements. In order to better compare the 14C and O2 GPP measurements we have updated our figures to include graphs with directly comparable units.

The authors' state that CO2 had no effect on bacterial production. However, looking at figure 7, there appears to be higher bacterial abundance between days 8 – 14 in the high CO2 treatments, which are not observed in bacterial productivity, indicating that bacterial production per cell is lower at high CO2? Surely, this is a CO2 response?

Response: The reviewer is correct that bacterial productivity appears to be lower at

high CO2. We determined that this result was likely an indirect effect of a decline in grazing pressure from heterotrophic nanoflagellates rather than a direct CO2 response on bacterial productivity. We have addressed this in the discussion (P18 Line 4-8).

The C:N data for POM is interesting but it is hard to discount carbon overconsumption without also looking at DOM. This would also be useful in interpreting the GPP 14C results. Respiration rates would also be useful. I realize these measurements can't be taken but the authors' should discuss these factors.

Response: We agree that in the absence of DOM analysis some carbon overconsumption cannot be excluded and thus, we have included an acknowledgement of this in our discussion. Respiration rates were measured and used to calculate GPP-O2. We refer to P15 Line 18-20 in the discussion where we address this concern.

In the methods section the authors' should justify the length of acclimation, why it was done under low light and why a blue filter was used.

Response: The CO2 acclimation was performed for 5 days as this timeframe has been commonly used in mesocosm experiments to reach high CO2 concentrations in a step-wise manner (e.g. Riebesell et al., 2013, Schulz et al., 2017). These studies used the naturally depleted nutrient concentrations at the start of their experiment to limit phytoplankton growth during acclimation. Instead, we were incubating water that was replete in nutrients and limited phytoplankton growth using low light. We used a blue filter over the lights in order to convert the tungsten lighting to a daylight spectral distribution. We have updated the methods section to include these justifications (please also see response 4 to reviewer 1).

References:

Riebesell, U., J. Czerny, K. von Bröckel, T. Boxhammer, J. Büdenbender, M. Deckelnick, M. Fischer, et al. 2013. "Technical Note: A Mobile Sea-Going Mesocosm System – New Opportunities for Ocean Change Research." Biogeosciences 10 (3). Coperni-

cus Publications: 1835–47. doi:10.5194/bg-10-1835-2013.

Schulz, Kai G, Lennart T Bach, Richard G. J. Bellerby, Rafael Bermúdez, Jan Büden-bender, Tim Boxhammer, Jan Czerny, et al. 2017. "Phytoplankton Blooms at Increasing Levels of Atmospheric Carbon Dioxide: Experimental Evidence for Negative Effects on Prymnesiophytes and Positive on Small Picoeukaryotes." Frontiers in Marine Science 4 (March): 64. doi:10.3389/fmars.2017.00064.
* * *
[Figure]

**Fig. 1.** CO2 threshold analysis for chlorophyll a accumulation, 14C-gross primary productivity rate, and accumulation of particulate organic nitrogen

---

## Author Response (AR1)

We thank the Editor for their recommendation for publication of our manuscript with minor revisions and have submitted an updated version that has addressed the reviewer comments.

[revised manuscript text omitted]